# Intraarticular Ligament Degeneration Is Interrelated with Cartilage and Bone Destruction in Osteoarthritis

**DOI:** 10.3390/cells8090990

**Published:** 2019-08-27

**Authors:** Gundula Schulze-Tanzil

**Affiliations:** Institute of Anatomy, Paracelsus Medical University (PMU), Salzburg and Nuremberg, Prof-Ernst Nathan Strasse 1, 90419 Nuremberg, Germany; gundula.schulze@pmu.ac.at; Tel.: +49-(0)-911-398-(11)-6772

**Keywords:** osteoarthritis, ligament, ACL, degeneration, synovitis, meniscotibial ligaments, enthesis, chondroid metaplasia, BMP signaling

## Abstract

Osteoarthritis (OA) induces inflammation and degeneration of all joint components including cartilage, joint capsule, bone and bone marrow, and ligaments. Particularly intraarticular ligaments, which connect the articulating bones such as the anterior cruciate ligament (ACL) and meniscotibial ligaments, fixing the fibrocartilaginous menisci to the tibial bone, are prone to the inflamed joint milieu in OA. However, the pathogenesis of ligament degeneration on the cellular level, most likely triggered by OA associated inflammation, remains poorly understood. Hence, this review sheds light into the intimate interrelation between ligament degeneration, synovitis, joint cartilage degradation, and dysbalanced subchondral bone remodeling. Various features of ligament degeneration accompanying joint cartilage degradation have been reported including chondroid metaplasia, cyst formation, heterotopic ossification, and mucoid and fatty degenerations. The entheses of ligaments, fixing ligaments to the subchondral bone, possibly influence the localization of subchondral bone lesions. The transforming growth factor (TGF)β/bone morphogenetic (BMP) pathway could present a link between degeneration of the osteochondral unit and ligaments with misrouted stem cell differentiation as one likely reason for ligament degeneration, but less studied pathways such as complement activation could also contribute to inflammation. Facilitation of OA progression by changed biomechanics of degenerated ligaments should be addressed in more detail in the future.

## 1. Introduction

Osteoarthritis (OA) is the most common joint disease. The central feature of OA is joint cartilage degradation and loss, radiologically visible by joint space narrowing. It is accompanied by signs of misrouted bone remodeling resulting in subchondral bone sclerosis, osteophyte development and biomechanical failure leading to subchondral bone cyst formation and cracks [1]. During OA, the biomechanics of the joint changes due to the loss of cartilage and metabolic disturbances leading to an altered subchondral bone elasticity and stiffness [2,3]. The influence of an altered biomechanics of joint-associated ligaments induced by OA on disease progression has not been further addressed. On the contrary, it is well known that joint instability caused by e.g., ligament laxity, ruptures, or proprioceptive deficits induces OA [4,5]. OA is driven by inflammatory flare ups associated with pro-inflammatory mediators and cytokines such as tumor necrosis factor (TNF)α, interleukin (IL)-1β, and IL-6 as the key players [6,7], activated complement [8,9], but also elevated TGFβ1 implicated in osteophyte formation [10].

It has also been shown that intraarticular ligaments directly respond to OA associated inflammation with degeneration [11,12,13]. The anterior and posterior cruciate ligaments (ACL, PCL) are intraarticularly localized and connect the inner surfaces of the lateral (ACL) and medial femoral condyles (PCL) with the anterior and posterior intercondylar areas of the tibia. Four meniscotibial ligaments unite the anterior and posterior horns of both menisci with the tibial plateau [14]. These intraarticular ligaments are directly neighbored to the osteochondral unit affected by OA of the knee. This unit comprises both the damaged joint cartilage and the destructed subchondral bone to which these ligaments are attached with their bony attachment zones (called entheses). Furthermore, they are in contact to the inflamed synovium and synovial fluid and hence are highly exposed to the inflammatory joint milieu in OA.

Joints represent a functional unit, which can be understood as an organ, and for this reason, OA is a whole joint disease [15]. This concept [15] is well accepted [16,17]. It describes that inflammation extends to all joint-associated structures including the joint capsule with the synovial membrane, intraarticular fat pads, menisci, and ligaments. Changes in the neighbored bone marrow such as edema, necrosis, bleeding, or fibrosis are also observed [18,19,20,21]. The largest joint of the body that is most often affected by OA is the knee, a condition designated as gonarthrosis. Hyperplasia of the synovial membrane, synovitis, and in some selected cases pannus formation can be observed in gonarthrosis [22]. In addition, menisci are affected by degradation and the Hoffa’s fat pad responses with the production of many inflammatory mediators [23]. Furthermore, diverse features of ligament degeneration were observed in OA [11,12].

Moreover, a ligament-centric concept of disease was proposed for early nodal OA, which represents a type of OA with multiple joint involvement [24]. Ligament entheses could contribute to joint inflammation in OA [25]. Since the central symptom of OA is pain, and joint cartilage is fully devoid of nerves, ligaments that contain pain receptors such as free nerve endings are likely involved in the pain sensation in OA, together with other innervated tissues of the joint such as the joint capsule with the synovium, subchondral bone, and surrounding connective tissues [26]. Hence, the consequences of ligament inflammation and degeneration during OA should attract deeper interest.

## 2. Anatomy of Intraarticular Ligaments of the Knee

Ligaments connect two bones or fibrocartilaginous structures with bone (for example the menisci) and exert stabilizing functions by balancing the tension between them. They show distinct differences to tendons, which link muscles with bone [27]. Moreover, they have also important proprioceptive functions containing various mechanoreceptors [28]. The loss of proprioceptive sensation in the ACL may favor the development of joint dysfunction and OA [4]. The ACL is involved in the sensomotoric response important for regular knee function and balanced loading [29]. Ligaments are bradytrophic tissues with poor blood supply and very low cell content but have an abundant extracellular matrix (ECM), which is responsible for their unique biomechanical behavior. In aged ligaments, the cell content can decrease further, predisposing them for degeneration or ruptures by additional reduction of their low intrinsic healing capacity. Mature ligaments contain mainly one cell type: differentiated ligamentocytes. Nevertheless, some less differentiated but synthetically more active ligamentoblasts can be detected in ligaments, particularly during maturation and probably also during remodeling [30]. In addition to ligamentocytes, ligamentoblasts, and minor numbers of other cell types including endothelial cells, fibroblasts, adipocytes, immune cells, and fibrochondrocytes, a reservoir of multipotent ligament stem/progenitor cells (LSPC) has been detected in ligaments [31]. These cell types might be directly involved in ligament tissue alterations during OA.

### 2.1. Anterior Cruciate Ligament (ACL)

The ACL (Figure 1) crosses the joint cavity of the knee. It is covered by the synovial membrane and hence, mostly protected from the direct contact with the synovial fluid (Figure 1A). It is characterized by a double-bundle structure consisting of an anteromedial and posterolateral bundle, which develop embryologically separately [32] and from the similar group of precursor cells (so-called blastem) like the menisci [33] (Figure 1B). Its tibial attachment site is localized in the immediate neighborhood to the infrapatellar fat pad (Hoffa’s fat pad, Figure 1A) and its femoral attachment is surrounded by some unnamed intraarticular fatty tissue within the intercondylar notch of the femur [34].

Its midsubstance possesses a hierarchical histological architecture: Multiple type I collagen fibrils form a collagen fiber that constitutes bundles enwrapped by a thin layer of flexible connective tissue named the endoligament (comparable to the endotenon in tendons). Fiber bundles and the endoligament generate a subfascicular unit. Several subfasciculi are embraced by another connective tissue sheet called the periligament (comparable to the peritenon in tendons), which is substantially thicker than the endoligament and forms fascicles. These are usually twisted in the ACL [34]. The ligament is surrounded by the epiligament, and then it is covered by the synovial layer [34,35] (Figure 1C). The synovium, especially near the attachment points of the ACL contains also many of the mechanoreceptors observed in the ACL represented by free nerve endings, Ruffini corpuscles, Pacinian corpuscles, and Golgi tendon bodies [28]. The ACL contains LSPCs in its core and synovium [31]. The tibial and femoral attachment zones of the ACL differ from the midsubstance histoarchitecture representing fibrocartilaginous entheses and the epiligament expression pattern can be distinguished from the fascicles and subfascicles [36]. The ACL differs in comparison to other ligaments by less compact collagen architecture, differences in cell nuclei shape, higher glycosaminoglycan and elastin contents, greater decorin, fibromodulin, transforming growth factor (TGF)β1, and hypoxia inducible factor 1 (HIF1α) gene expressions [36,37,38]. This highly organized histoarchitecture and characteristic expression profile is altered in degenerated ACLs (see 3.6).

### 2.2. Meniscotibial Ligaments

In a similar manner, the four meniscotibial ligaments (Figure 2) are localized within the joint cavity and hence are exposed to the inflammatory joint milieu in OA. These ligaments mediate an elastic fixation of the meniscal horns. They possess a fibrocartilaginous enthesis. The medial posterior meniscotibial ligament fixes the posterior horn of the medial meniscus and represents the localization of most often failure resulting in joint instability, which can facilitate OA development [14,39,40] and also induce cruciate ligament degeneration [40].

### 2.3. Blood Supply of the ACL and Meniscotibial Ligaments

Ligaments represent bradytrophic tissues that generally receive only poor blood supply. The low oxygen environment of the ACL is indicated by its relative higher HIFα expression compared with other ligaments [38]. ACL vascularization arises from the middle genicular and descending genicular arteries [34,41] and vessels of the Hoffa’s fat pad and adjacent synovium [34]. The above-mentioned arteries release periligamentous vessels. forming a network within the subintimal connective tissue layer of the synovial membrane (Figure 1). These periligamentous branches give rise to minor branches radially penetrating the ACL and joining the web of longitudinally aligned endoligamentous arterioles [41]. The inferior medial and lateral genicular arteries directly supply the distal part of the ACL with terminal branches. The ends of the ACL appear better vascularized than the midsubstance, and the proximal portion shows a greater vessel density than the distal portion [34]. Nevertheless, concerning this regional vascularization density some controversy exists [33]. In addition to its regular blood supply it is partly nourished by synovial fluid [42,43,44]. The outer fibrous zones of the menisci and their meniscotibial ligaments are physiologically supplied with blood from the middle, the lateral, and medial inferior genicular artery and their branches [45].

How dysregulated vascularization contributes to intraarticular ligament degeneration in OA remains unclear. It is known that angiogenesis is increased in other joint tissues affected by OA [46], particularly hypervascularization of the inflamed synovial membrane can be observed (Figure 3), which is also a source for ACL nutrition, but also blood vessels penetrating the osteochondral junction, the inner parts of the menisci, and osteophytes can be detected [47,48,49]. In addition, some degenerated ACLs show hypervascularization of the midsubstance (Figure 4).

## 3. Role of Intrinsic Stem Cells in Ligaments and the ACL

Stem cells have thoroughly been investigated in tendons. Since tendons and ligaments are very similar, one could probably translate known facts to ligaments. As in tendons, LSPCs could serve as a source of repair in injured ligaments [30]. However, for tendon stem/progenitor cells (TSPCs), it is suggested that they could also contribute to pathological features by differentiating into a non-tendon lineage such as fibrocartilage or facilitate scar formation [50,51,52]. Features such as fibrovascular scar, fatty tissue deposition, or heterotopic ossification are the results of misrouted TSPC differentiation in tendon healing [52].

Therefore, the question arises whether LSPCs contribute to the development of ligament degeneration [50]. There are several factors suspected to influence the capacity of these TSPCs to differentiate into the tenogenic lineage and, thereby, to support tendon healing such as the capacity for tenomodulin expression [52]. Different niches of TSPCs could be postulated, including a specialized perivascular niche [53,54,55]. Vascular stem cells have also been shown in the ACL [56]. Bi et al. reported that particular ECM components such as biglycan and fibromodulin are crucial for maintaining the stem cell niche milieu in tendon [57]. Despite not reported so far, inflammation and changed biomechanical conditions might also alter the LSPC niche.

TSPCs have a low immunogenicity, expressing only very low levels of MHC class II, CD86 and CD80 on their cell surface, which usually mediate a T-cell response and they do not induce complement-mediated cytotoxicity. Altogether, they present escape strategies for host T-cell and immune attack [58]. Stem cells are known to release trophic and immunomodulatory factors packaged into extracellular vesicles (EVs). Small microvesicles or even smaller exosomes contain diverse mediators such as cytokines and micro-RNAs as a mean of cell-cell communication, thereby exerting trophic effects on other cells [59]. EVs are released by articular chondrocytes and synoviocytes, whereby their amount and content changes in OA [60,61]. EVs are delivered into the synovial fluid and hence, might also reach intraarticular ligaments such as the ACL, PCL, and meniscotibial ligaments. However, the role of EVs for cell–cell communication has so far not been reported in these ligaments.

Stem cells such as mesenchymal stromal cells (MSC) can be found in the ACL as demonstrated in many studies [56,62,63,64,65]. Intrinsic stem cell subpopulations of the rabbit ACL could be localized in the sheath and core of the ACL as reported by Chen et colleagues [31]: The ACL sheath harbored vascular stem cells, whereas the core contained LSPCs. Both populations differed in regard to clonogenicity, proliferation activity, and multilineage differentiation capacity [31]. ACL-derived MSCs (ACL-MSCs) were characterized by high CD73 and CD90 expression. CD34 positive [56] and negative [66] stem cell populations have been shown in the ACL. Freshly isolated ACL-MSCs could be distinguished from bone marrow (BM)-MSCs in their differentiation capacity with high tendency to differentiate into the ligament lineage [67]. However, after long-term culturing, their cell surface expression profile became very similar. Since it is well known that OA is an age-associated disease, the question of whether ACL-MSCs from young and old donors differ becomes relevant. Gene expression associated with cytoskeleton and protein dephosphorylation, but also other pathways could be distinguished in ACL-MSCs from young and old donors [68]. The Hoffa’s fat pad and the synovial membrane covering the ACL are also known to contain stem cells [69,70]. Stem cells of the Hoffa’s fat pad change their expression pattern in OA [71]. In the synovial membrane, some Muse-like (stress enduring cluster forming multipotent) stem cells have been detected with strong chondrogenic potential expressing markers of Muse cells such as stage-specific antigen (SSEA)-3 and CD105 as well as pluripotency markers such as Nanog, octamer binding proteins (Oct) 3,4, and sex determining region Y-box 2 (Sox2) [70]. Taken together, the ACL contains different stem cell niches harboring heterogeneous stem cell populations. Their fate under the conditions of gonarthrosis should be investigated in detail.

### Inflammation and Stem Cells Commitment

When investigating stem cells from the bone marrow or adipose tissue, it has been suggested that the inflammatory milieu in the joint and inflammatory mediators elevated in OA might interfere with the commitment of MSCs [7,72,73]. Interestingly, de Sousa and colleagues observed that synovial fluid–derived cells isolated from the synovial fluid of non-OA patients were not able to differentiate into osteogenic, adipogenic, and chondrogenic lineages, whereas those from OA patients showed a typical trilineage differentiation capacity [74]. The differentiation capability of synovial membrane derived MSCs from the sheath of the cranial cruciate ligament (CrCL) of dogs differed dependent on the severity of inflammation in the joint, so the authors concluded that MSCs pre-activated by inflammatory factors might offer a method of improving the differentiation potency of these cells [75]. Therefore, the influence of inflammation might depend on the source of MSCs and the particular lineage addressed—for instance, chondrogenesis/osteogenesis versus ligamentogenesis. Unfortunately, only a few studies address the tenogenic/ligamentogenic lineage under inflammatory conditions. In this regard, Brandt et al. found a reduced capacity of adipose tissue–derived mesenchymal stromal cells (ASCs) to undergo tenogenic differentiation in the presence of inflammatory mediators expected under OA conditions such as interleukin (IL)-1β and tumor necrosis factor (TNF)α [73]. Taken together, one could hypothesize that LSPCs could undergo a non-ligamentogenic differentiation under OA conditions responsible for typical features of ligament degeneration.

## 4. Contribution of Synovial Fluid and Synovitis to Ligament Degeneration

OA is often associated with synovitis, which amplifies inflammation within the joint. Synovitis is especially localized in areas around the affected joint cartilage in OA [22]. In addition, in OA associated synovitis the composition of the synovial fluid (synovia) is changed [74,76]. Interestingly, the synovial fluid contains also a population of cells with stem cell character [74].

Generally, synovial fluid is a transudate of blood plasma released through the fenestrated endothelium of the capillaries within the synovium, modified by the synoviocytes, which are localized within the intimal and subintimal layer of the synovial membrane [77,78]. The number of synoviocytes type A (representing a specialized macrophage population) and type B (being the synovial fibroblasts) increased in OA [79]. The amount of the synovial fluid is also elevated in the OA joint leading to joint effusion, e.g., by increased blood vessel perfusion and capillary or synovial membrane permeability in response to inflammation [22]. It contains many inflammatory mediators [76] and a lower amount of functionally important synovial constituents such as hyaluronan and lubricin, important for joint cartilage lubrication and regular articular cartilage function [22]. The physiological exclusion limit of the blood-joint barrier for particles has been assumed to be 50 nm [80]. The blood–synovia/joint barrier normally mostly protects the ACL from direct contact with the synovial fluid [34]. However, it becomes weaker in OA compared to other physiological conditions. Interestingly, joint cartilage degradation experimentally induced by collagenase treatment led to an increased endothelial transmigratory capacity of synovial fibroblasts [81]. This is in agreement with the higher number of colony forming units observed when culturing OA-derived synovial fluid in comparison to healthy synovia [74]. Synovia nourishes the main part of the joint cartilage and parts of the ACL [42]. Therefore, the changed synovia composition in OA due to synovitis might also affect ACL tissue homeostasis.

To grade the severity of synovitis in OA, a synovitis score has been proposed by Krenn and coworkers [82,83,84] (Table 1). This synovitis score embraces typical features of synovitis and includes assessing the enlargement of the lining layer, cellular density of synovial stroma, and leukocyte infiltration by giving each a score of 0–3 points and sum them [83,84] (Figure 3). Due to the intimate interrelation of the ACL and the covering synovial layer [35] supplying it with blood, synovitis becomes highly relevant in ACL degeneration. Using the above mentioned score, we confirmed a significant correlation between synovitis and ACL degeneration in human samples harvested from OA and non-OA patients [13] that is in agreement with a recent study in dogs [85]. This is not surprising, since the vessels of the synovium directly nourish the ACL [86] and the blood synovia barrier is weakened in OA. However, the signaling pathways involved in this interplay remain unclear. Since the contribution of complement activation to synovitis has been suggested [22] complement split fragments could also affect ligaments.

In vitro co-culture experiments showed that the gene expression pattern of lysyl oxidase and matrix metalloproteinases (MMPs) in ACL cells was influenced by synovial fibroblasts in cocultures [87]. Synovial fluid had a stimulatory effect on ACL cell proliferation in vitro [88]. A mean of cell to cell communication by synovial fibroblasts with other joint associated cells, particularly under OA conditions, might be the release of extracellular vesicles (EVs) containing inflammatory mediators and miRNAs [61].

## 5. Ligament Histopathology in OA and Aging

Meanwhile, it is known that OA is associated with ACL degeneration [11,12,89]. However, the degeneration of the non-ruptured ACL is difficult to detect non-invasively, e.g., by magnetic resonance imaging (MRI), in vivo [90]. This results in the need for further technical optimization. Hence, most studies investigating ACL degeneration are based on histological results such as that of Cushner et al. [91]. In this study, 47% of the osteoarthritic patient group had moderate/marked ligament degeneration, whereas no control specimen showed similar changes. Seventy-two percent of the controls were considered normal, compared with only 26% of the osteoarthritic group [91]. Typical histopathological features of the ACL degeneration, which can be observed in OA, include chondroid and mucoid metaplasia, calcification, cyst formation, alterations of the cell content and phenotype, and the occurrence of myofibroblasts [12,13] (Figure 4). The degeneration was associated with increasing expression of the chondrogenic transcription factor Sox9 and impaired expression of the ligament marker scleraxis (SCX) [92], suggesting an enhanced chondrogenesis. In addition, the elastin content with freshly produced oxytalan fibers, which represent bundles of microfibrils, increased in degenerated CrCLs of dogs [93].

OA is an age-related disease, and the ACL and PCL show distinct age-associated degenerative changes such as collagen fiber disorganization, decreased cellularity, and impaired expression of the myofibroblast marker alpha-smooth muscle actin (α-SMA), which has been implicated in formation of the wavy crimping of collagen bundles in ligaments and the progenitor cell marker STRO-1 [11,12,13,94]. Furthermore, the number of various mechanoreceptors has been shown to decrease in the CrCL of rabbits with increasing age, suggesting that the important proprioceptive functions might be impaired [28]. Histopathological score values assessing ACL degeneration and ligament sheath inflammation increased with age [11]. Collagen fiber disintegration was an early and the most obvious alteration observed in ligament aging [11]. A macroscopically visible thinning of the ACL was observed with increasing age [95]. In addition to the above mentioned studies in human patients, in aging dogs, which are prone to CrCL degeneration, cell loss and chondroid metaplasia could be found in the CrCL and, to a lesser degree, in the caudal cruciate ligament (CaCL), but it is rarely found in other ligaments such as the medial collateral ligament [96].

Nevertheless, age- and degeneration-related ligament alterations share some similarities but can also display differences, e.g., in regard to MMP expression and cellularity, which increase in degenerated but not in aged ACLs, as well as the expression of chondrogenic markers that predominate in degenerated ACL, indicating areas of chondroid metaplasia [12].

A significant correlation between ACL degeneration and cartilage degeneration was observed in the medial compartment of the knee joint [11]. We found a correlation between ACL degeneration and synovitis but not with calcium deposits in human ACLs [13] or with the occurrence of α-SMA positive myofibroblasts [97]. Calcium deposits contain frequently calcium pyrophosphate dihydrate crystals [98] and represent probably an independent pathology, affecting often simultaneously other cartilaginous and ligamentous structures in the body [13].

Mucoid degeneration of the ACL was also strongly associated with the development of cartilage lesions [99,100]. It is a little-known and probably underdiagnosed entity [101,102]. Mucoid ACL degeneration has early been reported and was suspected to be associated with arthritic and synovial inflammation caused by acute or repetitive knee traumata [103]. In another study, lesions associated with mucoid degeneration included meniscal tears in 33 knees and chondral lesions of at least Outerbridge grade 2 in 56 knees of the investigated 80 knees [101]. Mucoid degeneration of the ACL coincides with changes in the PCL such as thickening, which remain mostly asymptomatic [104]. Nevertheless, mucoid degeneration of the ACL was also found to be associated with an increased posterior tibial slope (PTS) [105]. The patients with mucoid degeneration of the ACL had a greater mean PTS than matched control patients (13.5° versus 9.4°, *p* < 0.001) [105]. Accordingly, the results of another study led to the conclusion that an increased PTS after high tibial osteotomy resulted in ACL degeneration [106]. Interestingly, a comparable correlation between increased PTS and narrowing of the intercondylar notch and ACL degeneration (chondroid metaplasia) could recently be observed in dogs [107]. However, in another study in humans PTS seemed not to be directly connected with OA [108]. Nevertheless, ACL degeneration and abnormalities detected by MRI correlated with more severe degenerative changes in the joint than in patients with unchanged ACLs, likely because of the greater joint instability by functional impairment of the ACL [109]. The above-mentioned typical features of ACL degeneration are included in several helpful scoring systems summarized in Table 1B. An additional criteria might be hypervascularization of the ACL (detectable in Figure 4J–L), which is so far barely represented.

## 6. Involvement of the Ligament Enthesis in the OA Pathogenesis?

Ligaments and tendons seem to contribute to early OA as confirmed in human hand OA by MRI analyses. Imaging indicated that the localization of bone edema and erosions in early OA were influenced by affected ligaments [114]. This suggests that the tendon/ligament bone attachment sites, the enthesis, could play a central role. The enthesis is defined as the site of insertion of a tendon, ligament, fascia, or even joint capsule into bone [115]. It has important biomechanical functions bridging the interface between two materials—bone and ligament (dense connective tissue)—with unequal biomechanical properties. Different types of entheses exist—they can be ligamentous or fibrocartilaginous [116]. Intraarticular ligaments in the knee such as ACL (Figure 5), PCL, and meniscotibial ligaments possess a fibrocartilaginous enthesis [14]. They show a zonal architecture with a tidemark between the calcified and non-calcified fibrocartilage zones like joint cartilage. Entheses are often in the immediate neighborhood of the synovial membrane and joint cartilage [117]. This situation is also found in the cases of the ACL and PCL entheses [117]. Moreover, during human ACL development, the ACL is directly attached to the immature joint cartilage (femoral site) or perichondrium (tibial site) [118]. In view of an inflamed synovium, the Synovial-Enthesial-Complex has been postulated to contribute possibly to OA [24,114,117], triggering cartilage degradation. This hypothesis was supported by investigations of this complex in the PCL by Binks et al. [119]. Interestingly, the meniscal enthesis also underwent changes during OA development such as a loss of tidemark integrity, doubled tidemark, increased size of the mineralized zone but less bone mineral density, osteophyte formations, and microcracks and calcium deposits in the ligamentous zone [120]. The potential role of ligaments and the enthesis complex in generalized OA was pointed out by McGonagle et al. [24]. Nevertheless, OA-associated changes in the enthesis were barely studied due to being technically challenging and the fact that a validated scoring system is so far lacking. Nevertheless, published data concerning histopathological changes of the ACL enthesis could not be found.

## 7. Candidate Signaling Pathways Involved in Ligament Degeneration

One can therefore conclude that all parts of intraarticular ligaments including the enthesis are affected by OA. However, the aberrantly activated signaling pathways involved in ligament metaplasia in OA remain mainly unknown. Some candidates could be suspected to be strongly involved, as detailed below.

The main differentiation factors in ligaments, Mohawk (MKX) and SCX, were down-regulated in fibroblast-like cells in ACLs degenerated during OA [12,121]. In contrast, chondrocyte-like cells became detectable in increasing numbers, expressing the chondrogenic transcription factor Sox9. It is known that Sox9 is upregulated by TGFβ1 during chondrogenesis. In addition, more cells expressing Runx2 which is an osteogenic marker, appeared [12]. Hypercellularity of degenerated ACLs (Figure 4J–K) suggests cell proliferation of less differentiated cells. Hence, dedifferentiation and transdifferentiation are typical features of OA in intraarticular ligaments.

### 7.1. Transforming Growth Factor Superfamily and Ligament Degeneration

TGFβ1, a member of the TGF superfamily, is activated in the subchondral bone, e.g., in response to altered loading due to joint instability by loss of ACL function. Accordingly, its concentration is elevated in human OA [122]. Higher concentration of TGFβ1 could be shown in the synovial fluid from OA compared to non-OA patients [74]. TGFβ1 induced a misrouted subchondral MSC differentiation in an osteogenic direction associated with increased angiogenesis [122,123]. Moreover, its ectopic overexpression induced features of OA in mice and TGFβ1 inhibition could counteract cartilage degeneration [122,123]. TGFβ1, contributes to osteophyte formation by inducing endochondral ossification [124]. The ACL has a higher TGFβ1 gene expression compared with other ligaments correlating with a high proteoglycan content—it is known that proteoglycans capture TGFβ1 within the ECM [38]. During degeneration, the content of proteoglycans with associated glycosaminoglycans increases (Figure 4) [13]. This possibly makes the ACL sensitive to OA. Considering the fact that depletion of biglycan and fibromodulin, which are important components of the stem cell ECM niche in tendon, affected tenogenic differentiation of TSPCs and tendon formation in vivo by dysbalanced BMP signaling [57], one could assume that inflammatory conditions strongly increased the amounts of captured TGFβ1, which might also change the niche. Hence, this could induce stem cell commitment in a non-ligamentogenic lineage.

Osteophytes were observed in the menisceal enthesis of OA patients [120]. TGFβ1 can be used to induce chondrogenic differentiation of ligament-derived stem cells in vitro [97]. Hence, it could be suggested that this TGFβ-associated signaling pathway might contribute to chondroid metaplasia. TGFβ has been implicated in chondrocyte hypertrophy and subsequent ossification in OA, possibly contributing to osteophytes [10]. Accordingly, fine-tuned regulation of BMP signaling establishes the balance between chondrogenesis and fibrogenesis (ligamentogenesis) during joint development [125]. BMP2 and the down-stream signaling molecule of the canonical BMP pathway mothers against decapentaplegic homolog (SMAD)8 allows tenogenic differentiation of MSCs in vitro and in vivo [126]. In a previous study investigating the effects of another member of the BMP family, we found that, in cultured ACL ligamentocytes derived from OA patients, BMP2-mediated SMAD6 expression was significantly lower than in cells from donors free of OA at 4 h. SMAD6 is an inhibitor of BMP2 signaling. In addition, OA-ligamentocytes had a lower basal ID-1 expression [13]. BMP2 is known as a candidate to improve ACL integration into bone during healing [127]. In this regard, a set of miRNAs (e.g., let-7f-5p, miR-26b-5p and miR-146a-5p) was found to be upregulated in ACLs of OA patients compared with healthy individuals, while others were downregulated (e.g., miR-18a-3p, miR-138-5p, and miR-485-3p). Among others, the BMP2, TGFβ, and IL-6 genes were predicted as potential targets of these dysregulated miRNAs [128]. Hence, one can assume that a dysregulation of BMP family candidates involved in remodeling of joint cartilage, bone and ligaments by OA might also contribute to features of ligament degeneration.

### 7.2. Complement Cascade

In recent years, the involvement of complement activation has been strongly implicated in OA and synovitis [8,9,22,129]. Complement could also contribute to tendon degeneration since complement regulatory components are expressed in tenocytes, and the complement split fragment C3a elicited a catabolic response in tenocytes as shown by in vitro studies [130,131]. IL-6, which is present under OA condition but also the complement split fragment anaphylatoxin C3a, which is released during activation of the complement cascade, impaired the gene expression of cytoprotective complement regulatory proteins (CRP) in tenocytes. In addition, C3a stimulation lead to an up-regulation of TNFα and IL-1β mRNA in tenocytes. Degenerated tendons revealed an increased expression of the anaphylatoxin receptor C5aR and a reduced expression of the CRP CD55 [130]. However, this catabolic pathway has not been further addressed in the ACL so far.

## 8. Materials and Methods

Research was undertaken using Medline and Google scholar, using mesh terms such as ligament/enthesis degeneration associated with osteoarthritis; ligament degeneration and osteoarthritis/cartilage degradation; ligament degeneration and score; ACL and stem cells; mucoid degeneration of the ACL; synovitis and OA; misrouted stem cell differentiation; meniscotibial/meniscal ligaments/menisceal enthesis and OA; osteophytes and enthesis, TGFβ and OA, BMP pathway and OA, meniscotibial ligaments and OA; and others.

## 9. Conclusions

Intrarticular ligament degeneration is a common feature accompanying cartilage and bone degradation in OA. The inflammatory milieu in OA associated with the increase in diverse mediators and regulatory miRNAs might facilitate misrouted ligament stem cell commitment and ligamentocyte dedifferentiation (Figure 6). Chondroid metaplasia has been reported as a key feature in ligaments degenerated due to OA (Table 1). It is characterized by increased Sox9 expression, proteoglycan deposition and a dominating fibrochondrocytic cell phenotype. Dysregulated TGFβ/BMP signaling might substantially contribute to this feature. TGFβ1 in particular presents a link between the degeneration of ligaments and the osteochondral unit in OA [122,123]. Degeneration and chondroid metaplasia of the ACL and meniscotibial ligaments results in changed biomechanics, possibly impairing the stabilizing function of these ligaments and probably further accelerating OA progression (Figure 6). The intimate cross-talk between subchondral bone, cartilage alterations, and ligament degeneration should be investigated in more detail in the future.

## Figures and Tables

**Figure 1 cells-08-00990-f001:**
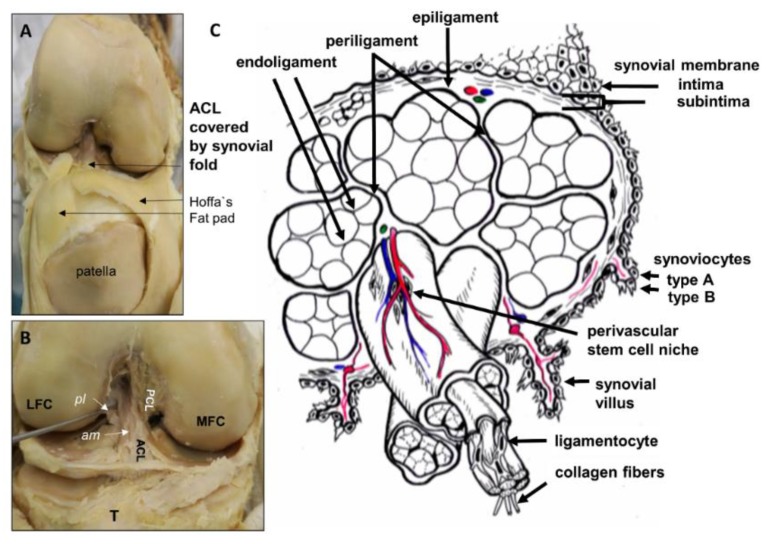
Human anterior cruciate ligament (ACL) gross anatomy and scheme of its histoarchitecture. (**A**): Gross anatomy of the ACL (visible within the intercondylar notch) covered by the synovial membrane and neighbored to the Hoffa’s fat pad. A frontal view of the knee joint with opened joint capsule is shown. (**B**): The synovial membrane has been removed and the anteromedial (am) and posterolateral (pl) bundles of the ACL and the posterior cruciate ligament (PCL) attachment site became visible. (**C**): Scheme of the micoanatomical hierarchical structure of the ACL midsubstance. Fascicles are surrounded by the periligament, subfascicles by the endoligament. LFC, MFC: lateral and medial femoral condyles, T: Tibia.

**Figure 2 cells-08-00990-f002:**
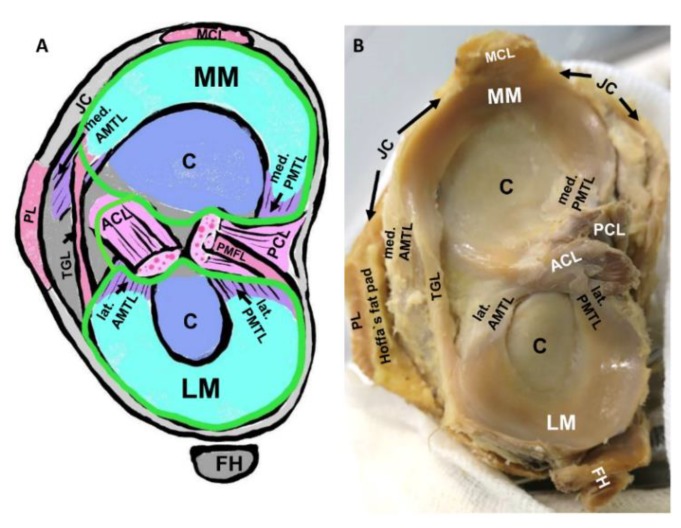
Intraarticular meniscotibial ligaments. (**A**): Schematic view on the tibial plateau. (**B**): Human gross anatomy of the tibial plateau with menisci and meniscotibial ligaments. ACL: anterior cruciate ligament, AMTL: anterior meniscotibial ligament, C: joint cartilage, FH: fibular head (in B covered by the popliteus muscle and the lateral collateral ligament), JC: joint capsule (gray), LM: lateral meniscus, MCL: medial collateral ligament, MM: medial meniscus, PCL: posterior cruciate ligament, PMFL: posterior meniscofemoral ligament, PMTL: posterior meniscotibial ligament, TGL: transverse genicular ligament. Green line: synovial membrane.

**Figure 3 cells-08-00990-f003:**
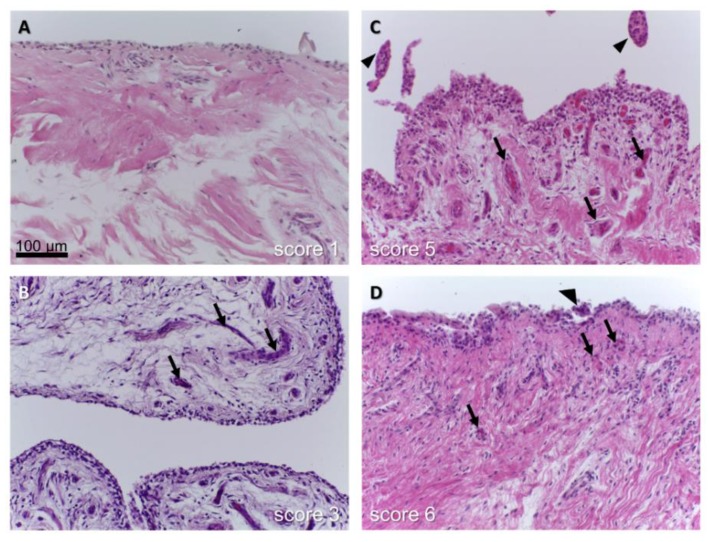
Synovitis of the synovial membrane covering the human ACL in osteoarthritis (OA). The score reported by Berger et al., was used. (**A**): Unchanged synovial sheath, (**B**): slightly, (**C**,**D**): inflamed synovium with hypervascularization. (arrows: vessels, arrow head: villus).

**Figure 4 cells-08-00990-f004:**
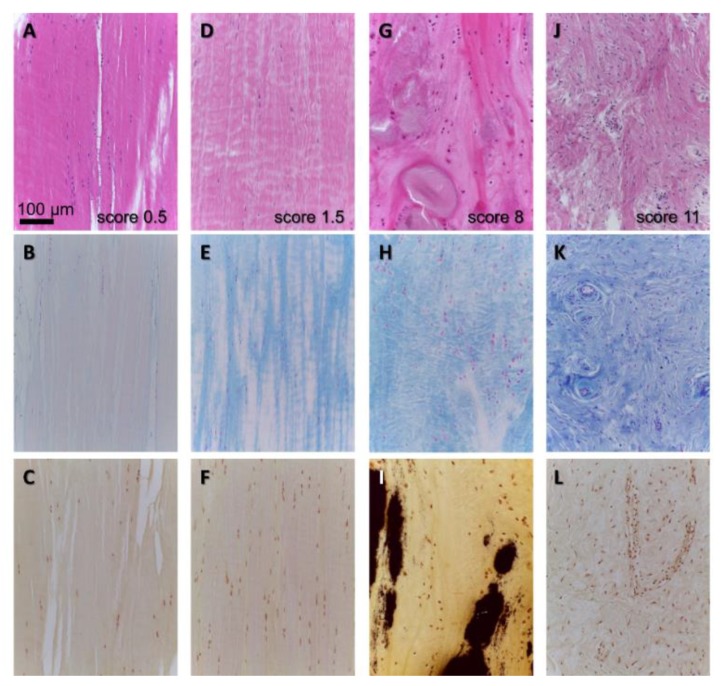
Histopathological features of human ACL degeneration in OA. (**A**,**D**,**G**,**J**): HE staining to give an overview over tissue organization. (**B**,**E**,**H**,**K**): Alcian blue staining to visualize distribution of sulfated glycosaminoglycans (blue), (**C**,**F**,**I**,**L**): van Kossa staining to detect calcium deposits (black). The scoring results with the scoring system used by [11] are depicted in (**A**,**D**,**G**,**J**), (**A**–**C**): nearly unchanged, (**G**–**I**): chondroid metaplasia and focal calcification, (**J**–**L**): ECM disintegration, hypercellularity, hypervascularization.

**Figure 5 cells-08-00990-f005:**
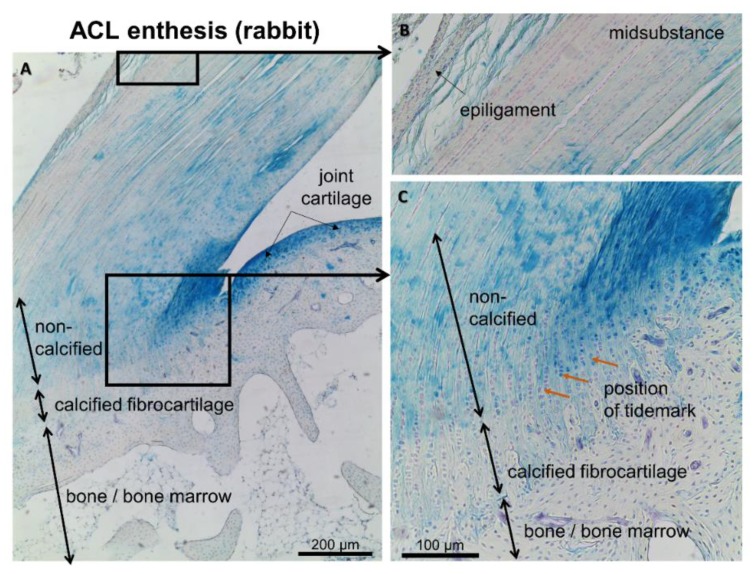
Enthesis of the unaffected rabbit ACL. Three different zones of the enthesis are shown using Alcian blue staining to depict sulfated glycosaminoglycan deposition (blue). (**A**): overview. (**B**): mid-substance. (**C**): enthesis zones.

**Figure 6 cells-08-00990-f006:**
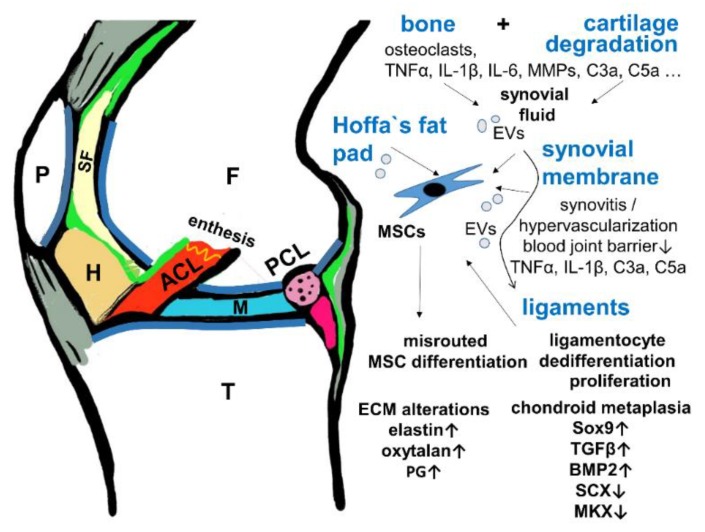
Hypothesis of the cross-talk between cartilage/bone and ligaments in OA. A scheme of a sagittal section of the knee is shown depicting the interrelation of the joint tissues involved in OA. ACL: red, F: femur, H: Hoffa’s fad pad, brown, M: meniscus, P: patella, PCL: transected, pink, SF: synovial fluid, pale yellow, T: tibia, green: synovial membrane, gray: patellar tendon, ligament and joint capsule. BMP2: Bone morphogenetic protein, EV: extracellular vesicles, PG: proteoglycans, MKX: Mohawk, SCX: scleraxis, TGFβ: transforming growth factor β.

**Table 1 cells-08-00990-t001:** Macroscopical and histological scores/key features applied to measure synovitis (**A**), degeneration of the ACL (**B**).

Score Details	Processing	References
**(A) Synovitis scores**		
three features of chronic synovitis (enlargement of lining cell layer, cellular density of synovial stroma, leukocyte infiltrates) (each from 0= absent to 3 = strong) Sum: 9 (0–1, no synovitis; 2–4, low-grade synovitis; 5–9, high-grade synovitis)	Histological score, human synovial membrane HE staining	[83]
**(B) Examples for cruciate ligament degeneration scores**		
1: normal, 2: abnormal (thinner than normal and sclerotic), 3: ruptured (complete disappearance of the ligament or persistence of only a few fibers)	Macroscopical score, human ACL	[110]
		[94]
		[11]
		[111]
1: normal ACL with no visible signs of disease, 2: moderately damaged or obvious disease such as the visible presence of fissuring, yet still overall an intact ACL, 3: complete rupture of the ACL	Macroscopical score, human ACL	[112]
0: normal, 1: degenerative lesions of collagen fibers in less than one third of the ligament thickness, 2: degenerative lesions of collagen fibers between one third and two thirds, 3: degenerative lesions of collagen fibers in more than two thirds	Histological score, human ACL longitudinal sections HE staining	[111]
abnormal PCL: -abrupt zones of loose fibrous connective tissue -cystic, myxoid, and/or mucoid alterations. Cystic degeneration: acellular cysts, myxoid pattern: replacement of the normal collagen pattern by spindle and stellate-type cells. Mucinous degeneration: pools of mucinous-like tissue. Categories: normal, slight, mild, moderate, and marked	Histological score, human PCL cross section of the tibial and femoral sites, two (lateral and medial) paramedian longitudinal sections HE staining	[98]
1: inflammation in the ACL, 2: mucoid degeneration, 3: chondroid metaplasia, 4: cystic changes, 5: orientation of collagen fibers changes were scored as follows: 0: no changes, 0.5: minimal changes, 1: mild changes, 2: moderate changes, 3: severe changes, sum: 0–15	Histological score, human ACL transversely and longitudinally (median) from the proximal 1/3 of the midsubstance and femur attachment site, HE staining (+Alcian blue staining)	[89] Was slightly modified and used by others: [11,13,94]
1: ligamentous structure with parallel bundles of closely packed collagen fibers, low cellularity with well spaced parallel fibroblasts showing extremely elongated nuclei: no secondary degenerative features 2: parallel arrangement of collagen and fibroblasts still detectable but some loss of compaction of collagen fibers and/or increased fibroblast nuclei. Nuclei are plumper, although still elongated. A few secondary degenerative features may be present 3: disorganization and disruption of the parallel arrangement of collagen fibers is more pronounced. Increased cellularity is often apparent and the nuclei may show loss of their bipolar nature. Secondary degenerative changes, such as calcification, ossification, cyst formation and myxoid changes may be prominent	Histological score, human ACL longitudinal sections HE staining	[112]
chondroid metaplasia (e.g., loss of spindle-shaped fibroblasts with transformation into ovoid or round nuclei with normal or clonal appearance; formation of perinuclear “halo” areas) and matrix degradation (e.g., loss of typical collagen fiber organization; increase in proteoglycans; woven collagen fiber, and fiber disorganization, Chondroid metaplasia and matrix degradation were graded: 1: no changes, 2: mild changes in a small area, 3: moderate, 4: severe, 5: severe diffuse changes	Histological score, canine CrCL longitudinal sections HE staining Alcian blue staining	[107,113]
(modified) Vasseur score grades 0–3: with several subcategories Grad 1: mild degenerative changes affecting the ligamentocytes and collagen bundles, small solitary and multiple areas of degeneration in the central region. Grade 2: large acellular areas, chondroid metaplasia, focal cell proliferation, collagen fibrils not within dense primary bundles, loss of typical bundling pattern Grade 3: a variety of severe degenerative changes (more than half of the ligament diameter involved) with tearing of deteriorated axial fibers.	Histological score, canine CrCL longitudinal sections HE staining	[93,96]
microfibril staining, increased staining of interfascicular and interbundle regions, ligament substance (intrabundle), extent and degree pericellular staining, each category up to 2 points, sum: 0–10	Microfibril staining, canine CrCL and CaCL, longitudinal sections Miller’s stain: elastin, oxytalan fibers	“Miller’s score” [93]
calcium distribution (none, single, multiple foci, throughout), quality of deposits (none, diffuse, compact), intensity of staining/color (none, low = brown, pronounced: brown + black, intense = black), sum: 0–8	Calcium deposition Van Kossa score, longitudinal sections Van Kossa staining	supplemental Table 1 in: [13]

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
