# Peer review of "Intraarticular Ligament Degeneration Is Interrelated with Cartilage and Bone Destruction in Osteoarthritis"

_cells, 2019, doi:10.3390/cells8090990_

Round 1

Reviewer 1 Report

This review gives an important and well-researched overview on the relationship of OA and ligament pathology. It is supported by several well-designed figures.

I only have a few suggestions to further improve the quality of the submitted manuscript:

I believe the manuscript would benefit from proof-reading by a native English speaking colleague, especially with regard to punctuation The subsections might be titled differently to improve the structure of the article. My rationale for the following suggestion is that basically, the whole article deals with ligament pathology, thus the subsection “ligaments” (now 3.) might make the structure more complicated than it actually is. My suggestion would be to have subsection titles as follows:
1. Introduction
(2. Osteoarthritis affects the whole joint) à integrate in introduction?
2. Anatomy of intraarticular ligaments (of the knee)
2.1 ACL
2.2 Meniscotibial ligaments
2.3 Blood supply of the ACL and meniscotibial ligaments
3. Role of intrinsic stem cells in ligaments
3.1 Inflammation and stem cells commitment
4. Contribution of synovial fluid and synovitis to ligament degeneration
5. Ligament histopathology in OA and aging
6. Involvement of the ligament enthesis in OA pathogenesis
7. Candidate signaling pathways involved in OA and ligament degeneration
7.1 TGFbeta family growth factors (also see comment below!)
7.2 Complement cascade
8. Materials and methods
9. Conclusions ad section 4/ 4.1: To the best of my knowledge, BMPs are part of the TGFB superfamily, not the other way around as stated here- this should be corrected throughout this section Mohawk is typically abbreviated MKX, not MHW (fig. 6)

Author Response

Intraarticular ligament degeneration accompanying cartilage and bone destruction in osteoarthritis

Gundula Schulze-Tanzil

The author would like to thank the reviewer for carefully reading the manuscript and their constructive comments. I modified the manuscript according to their suggestions and comments. A list of changes is reported below point by point. All changes performed are indicated in red and underlined in the revised version of the manuscript. Please refer to my point by point reply below. I hope that the revised manuscript is suitable now for publication in “Cells.

Please do not hesitate to contact me anytime for questions regarding this manuscript.

Sincerely,

Univ.-Prof. Dr. Gundula Schulze-Tanzil

Point by point reply:

Reviewer 1:

This review gives an important and well-researched overview on the relationship of OA and ligament pathology. It is supported by several well-designed figures.

I only have a few suggestions to further improve the quality of the submitted manuscript:

I believe the manuscript would benefit from proof-reading by a native English speaking colleague, especially with regard to punctuation.

Response: The manuscript was revised in regard to English style. The punctuation was checked.

The subsections might be titled differently to improve the structure of the article. My rationale for the following suggestion is that basically, the whole article deals with ligament pathology, thus the subsection “ligaments” (now 3.) might make the structure more complicated than it actually is. My suggestion would be to have subsection titles as follows:
1. Introduction
(2. Osteoarthritis affects the whole joint) à integrate in introduction?

Response: This paragraph was integrated into the introduction section as recommended.

Anatomy of intraarticular ligaments (of the knee)
2.1 ACL
2.2 Meniscotibial ligaments
2.3 Blood supply of the ACL and meniscotibial ligaments
3. Role of intrinsic stem cells in ligaments
3.1 Inflammation and stem cells commitment
4. Contribution of synovial fluid and synovitis to ligament degeneration
5. Ligament histopathology in OA and aging
6. Involvement of the ligament enthesis in OA pathogenesis
7. Candidate signaling pathways involved in OA and ligament degeneration
7.1 TGFbeta family growth factors (also see comment below!)
7.2 Complement cascade
8. Materials and methods
9. Conclusions ad section

Response: I thank the reviewer fort he advise and followed the order proposed by the reviewer. I did not move the conclusion section since it is placed here in the template.

4/ 4.1: To the best of my knowledge, BMPs are part of the TGFB superfamily, not the other way around as stated here- this should be corrected throughout this section Mohawk is typically abbreviated MKX, not MHW (Fig. 6)

Response: I followed the reviewer and used the proposed regular abbreviation and corrected the pathway.

Reviewer 2 Report

The author elucidated the importance of ligaments in osteoarthritis development. The role and the relationship between degenerated ligaments and destructed cartilage/bone during osteoarthritis were introduced. Moreover, the author also described important mediators, including stem cells, TGF-b/BMPs...to explain possible cross-talk between ligaments and cartilage/bone in OA. I think this is a comprehensive review article, which provides an organization and impact for readers to further understand the role of ligaments in OA development. The minor suggestion of mine is that the abbreviation should not be used when they appear in the first time in the context, for example, "ACL" in the abstract and "OA" in the introduction...etc. 

Author Response

Intraarticular ligament degeneration accompanying cartilage and bone destruction in osteoarthritis

 Gundula Schulze-Tanzil

 The author would like to thank the reviewer for carefully reading the manuscript and their constructive comments. I modified the manuscript according to their suggestions and comments. A list of changes is reported below point by point. All changes performed are indicated in red and underlined in the revised version of the manuscript. Please refer to my point by point reply below. I hope that the revised manuscript is suitable now for publication in “Cells.

Please do not hesitate to contact me anytime for questions regarding this manuscript.

Sincerely,

Univ.-Prof. Dr. Gundula Schulze-Tanzil

The minor suggestion of mine is that the abbreviation should not be used when they appear in the first time in the context, for example, "ACL" in the abstract and "OA" in the introduction...etc. 

Response: I inserted the explanations of the abbreviations.

Further revisions: scale bars were inserted in the figures now. Fig. 2B was added to the scheme (A) to better visualize the natural conditions of the meniscotibial ligaments in situ.
